# scRNA-Seq of Cultured Human Amniotic Fluid from Fetuses with Spina Bifida Reveals the Origin and Heterogeneity of the Cellular Content

**DOI:** 10.3390/cells12121577

**Published:** 2023-06-07

**Authors:** Athanasia Dasargyri, Daymé González Rodríguez, Hubert Rehrauer, Ernst Reichmann, Thomas Biedermann, Ueli Moehrlen

**Affiliations:** 1Tissue Biology Research Unit, Department of Surgery, University Children’s Hospital Zurich, 8032 Zurich, Switzerland; 2Functional Genomics Center Zurich, ETH Zurich and University of Zurich, 8057 Zurich, Switzerland; 3Faculty of Medicine, University of Zurich, 8006 Zurich, Switzerland; 4Zurich Center for Fetal Diagnosis and Therapy, University of Zurich, 8006 Zurich, Switzerland; 5Pediatric Surgery, University Children’s Hospital Zurich, 8032 Zurich, Switzerland

**Keywords:** amniotic fluid cells, single-cell RNA sequencing, spina bifida, amniotic fluid stem cells

## Abstract

Amniotic fluid has been proposed as an easily available source of cells for numerous applications in regenerative medicine and tissue engineering. The use of amniotic fluid cells in biomedical applications necessitates their unequivocal characterization; however, the exact cellular composition of amniotic fluid and the precise tissue origins of these cells remain largely unclear. Using cells cultured from the human amniotic fluid of fetuses with spina bifida aperta and of a healthy fetus, we performed single-cell RNA sequencing to characterize the tissue origin and marker expression of cultured amniotic fluid cells at the single-cell level. Our analysis revealed nine different cell types of stromal, epithelial and immune cell phenotypes, and from various fetal tissue origins, demonstrating the heterogeneity of the cultured amniotic fluid cell population at a single-cell resolution. It also identified cell types of neural origin in amniotic fluid from fetuses with spina bifida aperta. Our data provide a comprehensive list of markers for the characterization of the various progenitor and terminally differentiated cell types in cultured amniotic fluid. This study highlights the relevance of single-cell analysis approaches for the characterization of amniotic fluid cells in order to harness their full potential in biomedical research and clinical applications.

## 1. Introduction

Amniotic fluid (AF) is the liquid surrounding the fetus in the amniotic sac during gestation, and it serves for protecting the fetus and providing nutrients for its growth in addition to supporting the normal development of its gastrointestinal, respiratory, urinary and musculoskeletal system [1]. AF is comprised of water, electrolytes, carbohydrates, proteins and peptides, lipids, hormones [1] and cells [2]. Its complex composition fluctuates with gestation age and is a result of several processes during fetal development. Specifically in the second trimester, the processes affecting AF volume and composition include primarily fetal urination, and—to a lesser extent—fetal swallowing, fetal lung secretions and the transfer between AF and fetal blood (intramembranous pathway) and between AF and maternal blood (transmembranous flow) [3].

The cellular content of AF is of great interest, and not only has it been exploited for the prenatal diagnosis of genetic disorders [4,5], but it has also been evaluated preclinically as a source of cells for tissue engineering and regenerative medicine applications [6,7]. The cellular content of second-trimester AF consists predominantly of non-viable cells [8], with the viable cells constituting approximately 10% of the total cellular count [9] (20,000 viable cells per mL on average [10]). Among these, plastic-adherent colony-forming cells exist (<3% of total cell number [8]), and can be expanded in culture. The cellular content of AF is a mixture of heterogeneous populations resulting from the diverse processes affecting AF composition and the various exfoliating fetal tissues that are in contact with AF. In addition to the heterogeneity of cell types in each AF culture, the number and composition of viable AF cells vary among subjects and with gestational age [8]. Furthermore, congenital malformations can also influence the cellular content of AF. For instance, neural tube defects contribute neural cells [11], and abdominal wall abnormalities shed peritoneal cells in AF [8,12].

Interestingly, the presence of AF cells with various degrees of stemness has been proposed, and this has given rise to numerous research efforts in tissue engineering using AF as the cellular source [13]. Several reports have attributed features of pluripotency to cultured AF cells under standard culture conditions (namely, without culture factors that support pluripotency), including high telomerase activity [14], the embryonic stem cell markers Oct-4 [15,16], Nanog [17] and Sox2 [13,18], and the hematopoietic and embryonic stem cell marker c-Kit [19,20,21,22]. The last has been used for the isolation of amniotic fluid stem cells (AFSCs) that have been deemed pluripotent [21] or broadly multipotent [19]. The presence of amniotic fluid mesenchymal stem cells (AFMSCs) [23,24] has also been proposed, following the observation that mesenchymal cells exist among expanded AF cells and can be isolated by mechanical separation and selection with culture media [25]. Both AFSCs and AFMSCs have been shown to differentiate into lineages from the three germ layers [13,26,27]. Recently, however, the expression of pluripotency markers by AF cells has been questioned [28].

To harness the full potential of AF cells as a cellular source for tissue engineering, a complete characterization of their cellular content is indispensable. However, the tissue origin and lineage of the various populations within AF cell cultures have never been thoroughly characterized. Efforts thus far to characterize the viable adherent AF cells have focused on morphological, biochemical, molecular and phenotypical approaches [12,29,30,31]. Early classifications based on morphological and cell growth criteria suggested the presence of two (spindle-shaped and round-shaped [28,32]) or three (epithelioid, fibroblastic and amniotic [12,33]) cell types. Other reports attempting to characterize AF cells have shown the expression of mesenchymal [23] and epithelial markers [34,35], but also of neural [30,35], hematopoietic [36] and stem cell markers [30], by means of immunophenotyping and mRNA expression assays. Proteomics and transcriptomics analyses have further been employed to study marker expression by AF cells [16,31,34,37]. However, such bulk approaches cannot detect gene expression variability among subpopulations or individual cells and are not sufficient for the adequate characterization of each cell population given the polyclonal nature of AF cells in culture [29,35,36,38]. The identification of all cell types in AF requires approaches that enable the characterization of gene expression at a single-cell resolution.

In this study, we employed single-cell RNA sequencing (scRNA-seq) to address the heterogeneity of cultured AF cells of 20–24-week gestational age. Having access to AF from open fetal surgery for the prenatal repair of spina bifida aperta (SBA), we used four samples from fetuses with SBA. We also included a sample from a healthy fetus. Using differential gene expression analysis, we identified for the first time the cell lineage and tissue origin of all cell populations in cultured AF. We also examined the expression of pluripotency markers in AF cells and interrogated the presence of stem cell populations. Our analysis did not detect populations with pluripotency characteristics. However, we identified several cell types with potential utility in biomedical applications, such as regenerative medicine and drug screening. Our study provides a comprehensive list of markers for the characterization of the cell populations found in cultured AF that can better inform future research and therapeutic strategies using AF cells.

## 2. Materials and Methods

### 2.1. Amniotic Fluid Samples

Samples of AF (22–90 mL) from fetuses with spina bifida aperta were obtained during open fetal surgery for the prenatal repair of myelomeningocele performed at the University Children’s Hospital Zurich (gestational age 23–26 weeks). After collection, AF was stored at 4 °C until cell isolation (within 48 h). The AF sample from a healthy fetus (5 mL) was obtained from amniocentesis (gestational age 20.9 weeks). The gestational age of the normal fetus could not be matched precisely because amniocentesis is performed at <22 weeks and open fetal surgery is performed at 23–26 weeks.

### 2.2. Isolation and Culture of Amniotic Fluid Cells for Single-Cell RNA Sequencing

Five samples of amniotic fluid were cultured to obtain adherent cells (4 samples from fetuses with spina bifida aperta with mean gestational age 24.4 ± 0.4 weeks, and 1 from a healthy fetus at gestational age 20.9 weeks; see information on individual samples in Appendix A). Samples were used within 24–48 h from collection. AF was centrifuged (5 min, 300× *g*) and the supernatant was removed. The cells were resuspended in 10 mL Chang C medium (Irvine Scientific, Santa Ana, CA, USA) supplemented with 2 mM L-glutamine (Gibco, Waltham, MA, USA) and 1% penicillin/streptomycin (Gibco), and plated in a 10 cm petri dish. The AF cultures were left undisturbed to adhere for 5 days under a humidified atmosphere at 37 °C with 5% CO_2_ and then were washed 3 times gently with PBS to remove debris and non-adherent cells. The medium was replaced with fresh Chang C complete medium and the cells were kept in culture at passage 0 (P0) until large colonies appeared.

### 2.3. Single-Cell RNA Sequencing

For single-cell transcriptomics analysis of cultured AF cells, the droplet-based 10X Genomics Chromium scRNA-seq platform was used [39] at the Functional Genomics Center Zurich (University of Zurich, Zurich, Switzerland). In detail, when large colonies of AF cells appeared at P0, the cells were detached with trypsin/EDTA (Gibco) for 5 min and resuspended in Chang C complete medium supplemented with 0.5% BSA (*w*/*v*) (Gibco) at a concentration of 1000 cells per µL. The quality and concentration of the single-cell preparations were evaluated using a hemocytometer. Samples were immediately loaded into the 10X Genomics Chromium controller (Pleasanton, CA, USA). Approximately 17,000 cells were loaded per sample. The 10X Genomics libraries were prepared according to the manufacturer’s instructions (10X Genomics protocol Single Cell 3′ v3). Libraries were sequenced in an Illumina NovaSeq sequencer (Illumina Inc., San Diego, CA, USA) according to the 10X Genomics recommendations to a depth of around 50,000 reads per cell.

### 2.4. Single-Cell RNA-Seq Data Analysis

#### 2.4.1. Analysis of Each Sample

Reads were demultiplexed using the cell barcodes then aligned to the Homo Sapiens reference sequence (build GRCh38.p13) taken from Ensembl, and finally, the UMIs were collapsed and used for quantification. Empty droplets or droplets containing mostly ambient RNA were filtered. All the previous steps were performed using the CellRanger software (v3.1.0, 10XGenomics). Doublets were also discarded from further analysis using the R package scDblFinder [40]. Low-quality cells were identified based on the number of reads and genes and amount of mitochondrial content using the R package scater [41]. Cells that were outliers according to at least one of the following thresholds were excluded: number of reads < 3 MADs, number of gene < 3 MADs, mitochondrial content > 3 MADs. Additionally, we discarded genes with <1 UMI count in <0.01 of the remaining cells. The SCTransform [42] method from the R package Seurat [43,44,45] was used to normalize and scale the data.

#### 2.4.2. Integration of Multiple Samples

Multiple samples were combined using the Seurat integration workflow [43]. After integration, the dimensionality of the data was reduced using Principal Component Analysis (PCA). The main 30 principal components were used to perform unsupervised clustering with a resolution value of 0.4. After clustering and visualization with t-distributed stochastic neighbor embedding (t-SNE), positive markers that defined clusters compared to all other cells via differential expression were found. A logistic regression framework including the cell cycle as a latent variable was used. Genes with an average at least 0.25-fold difference (log-scale) between the cells in the tested cluster and the rest of the cells and an adjusted *p*-value < 0.05 were declared as significant. Cell-cycle phases were predicted using a function included in the scran R package [46] that scores each cell based on expression of canonical marker genes for S and G2/M phases. We computed aggregated expression scores for the neural markers using the package decoupleR [47].

### 2.5. Data Access

Our data were deposited in the NCBI Gene Expression Omnibus (GEO) under accession number GSE206696.

### 2.6. Gene Ontology Enrichment Analysis (GO Analysis)

The enrichment analysis was performed with the R package clusterProfiler [48], taking the positive markers per cluster and using the Biological processes from Gene Ontology as categories. GO terms with an adjusted *p*-value smaller than 0.05 were considered significant, and were depicted and ranked by adjusted *p*-value.

### 2.7. Fluorescence-Activated and Magnetic Cell Sorting of c-Kit-Positive Cells

Isolation of c-Kit+ AF cells was performed as described before [22]. Amniotic fluid samples from fetuses with spina bifida aperta (gestational age 23–26 weeks) were centrifuged (5 min, 300× *g*) and the supernatant was removed. The cells were resuspended in Chang C medium (Irvine Scientific) supplemented with 2 mM L-glutamine and 1% penicillin/streptomycin, and plated over glass coverslips in 35 mm petri dishes. Medium was changed after 5 days. When large colonies appeared (P0), adherent cells were washed twice with PBS and detached with trypsin/EDTA, and were then subjected to fluorescence-activated cell sorting (FACS) or magnetic-activated cell sorting (MACS) for the isolation of c-Kit+ cells.

For the isolation of c-Kit+ cells by MACS, the human CD117 MicroBead Kit with MS columns and a MiniMACS Separator (all from (Miltenyi Biotec, Bergisch-Gladbach, Germany) were used according to the manufacturer’s instructions and the published protocol [22]. For FACS sorting of c-Kit+ cells [20], detached AF cells were stained with Zombie NIR (BioLegend, San Diego, CA, USA) and CD117-PE antibody (clone 104D2, Dako, Agilent, Santa Clara, CA, USA), and c-Kit+ cells were sorted using a BD FACSAria III equipped with the FACSDiva Software (BD Biosciences, San Jose, CA, USA) at the Cytometry Facility of the University of Zurich using a 100 μm nozzle. Sorted cells by MACS or FACS were plated in MEM Alpha (Gibco) containing 20% Chang C medium, 15% FBS (Gibco), 2 mM L-glutamine and 1% penicillin/streptomycin.

### 2.8. Cytospin

AF cultures at P0 from fetuses with spina bifida aperta, grown in 10 cm petri dishes for 5–7 days to 70–80% confluency, were detached with trypsin/EDTA and then spun onto glass slides using a Shandon Cytospin 4 cytocentrifuge (Thermo Fisher Scientific, Waltham, MA, USA) and Shandon EZ Double Cytofunnels, according to the manufacturer’s instructions. Cells on glass slides were fixed for 5 min in cold (−20 °C) acetone/methanol (50:50), air dried and washed in PBS before proceeding to immunofluorescence staining. AF cells after c-Kit+ selection by MACS were also spun onto glass slides with the same procedure and prepared for immunostaining to evaluate the purity of the isolated cell populations by c-Kit immunostaining.

### 2.9. Immunofluorescence Staining for Microscopy

AF cell cultures at P0 from fetuses with spina bifida aperta (isolated as described above) were grown in 10 cm petri dishes for 5–7 days to 70–80% confluency, and subsequently fixed for 5 min in cold (−20 °C) acetone/methanol (50/50), air dried and rehydrated in PBS. Cells were blocked for 30 min with PBS containing 2% BSA(blocking buffer) (Sigma-Aldrich, Buchs, Switzerland) and then incubated in primary antibody for 1 h at RT. Primary antibodies were diluted in blocking buffer: mouse anti-human pan-cytokeratin antibody (1:100, clone AE1/AE3, Dako), rabbit anti-human cytokeratin 18 antibody (1:50, Abcam, Cambridge, UK) and rabbit anti-human cytokeratin 19 antibody (1:50, Abcam). Cells were washed 3× with PBS at RT and then incubated in secondary antibody for 1 h at RT. Secondary antibodies were diluted in blocking buffer: donkey anti-mouse Alexa Fluor 488 (1:400, Abcam), goat anti-rabbit Alexa Fluor 568 (1:400, Abcam) and donkey anti-rabbit Alexa Fluor 488 (1:400, Abcam). For negative control samples, the cells were stained with the secondary antibody alone (without primary antibody). Cells were washed 3× in PBS for 10 min at RT, then mounted with a coverslip using medium Fluoroshield with DAPI (Sigma-Aldrich).

Fixed cytospin preparations of AF cultures from fetuses with spina bifida aperta (prepared as described above) were blocked with PBS containing 2% BSA (blocking buffer) and then incubated with primary antibody diluted in blocking buffer for 1 h at RT. Primary antibodies were diluted in blocking buffer: mouse anti-human EpCAM antibody (1:200, clone VU-1D9, Abcam), mouse anti-human CD90 antibody (1:50, Dianova, Hamburg, Germany), rabbit anti-human cytokeratin 18 antibody (1:50) and rabbit anti-human cytokeratin 19 antibody (1:50). Cells were washed 3× with PBS at RT and incubated in secondary antibody for 1 h at RT. Secondary antibodies were diluted in blocking buffer: donkey anti-mouse Alexa Fluor 488 (1:400, Abcam), goat anti-rabbit Alexa Fluor 568 (1:400, Abcam), donkey anti-rabbit Alexa Fluor 488 (1:400, Abcam) and donkey anti-mouse Alexa Fluor 568 (1:400, Abcam). For negative control samples, the cells were stained with the secondary antibody alone (without primary antibody). Cells were washed and mounted with a coverslip as described earlier.

Fixed cytospin preparations of AF cultures selected for c-Kit by MACS at P0 (procedure described above) were blocked with PBS containing 2% BSA and then incubated with mouse anti-human c-Kit antibody (Ab81, Cell Signaling Technology, Danvers, MA, USA) diluted 1:50 in blocking buffer for 1 h at RT. Cells were washed 3× with PBS at RT and then incubated with donkey anti-mouse Alexa Fluor 488 secondary antibody (Abcam) diluted 1:400 in blocking buffer for 1 h at RT. After PBS washing (3×), the samples were mounted with a coverslip using medium Fluoroshield with DAPI.

Pictures of immunofluorescence staining were taken with a DXM1200F digital camera connected to a Nikon Eclipse TE2000-U inverted microscope. The device is equipped with Hoechst 33342-, FITC- and TRITC-filter sets (Nikon AG, Switzerland; Software: NIS-Elements).

### 2.10. Flow Cytometry Analysis of Mesenchymal Stem Cell Markers 

AF cell cultures at P0 from fetuses with spina bifida aperta (isolated as described above), grown in 10 cm petri dishes for 5–7 days to 70–80% confluency, were detached with Accutase (StemCell Technologies, Vancouver, BC, Canada) and resuspended in staining buffer (PBS supplemented with 0.5% BSA (*w*/*v*) and 2 mM EDTA). Cells were stained with primary antibodies for 30 min at 4 °C in staining buffer: anti-human CD29-Alexa Fluor 488 antibody (1:50, clone TS2/16, BioLegend), anti-human CD73-PE antibody (1:50, clone AD2, BD Biosciences, Eysins, Switzerland), anti-human CD44-PE antibody (1:50, clone G44-26, BD Biosciences), anti-human CD90-FITC (1:50, clone 5E10, BioLegend) and anti-human CD105-Alexa Fluor 488 antibody (1:50, clone 43A3, BioLegend). Samples were analyzed on a BD LSRFortessa flow cytometer equipped with the FACSDiva Software at the Cytometry Facility of the University of Zurich. Data were analyzed using FlowJo (BD Life Sciences, Ashland, OR, USA). 

## 3. Results

### 3.1. Single-Cell Transcriptomic Analysis Identifies Nine Cell Types in Cultured Amniotic Fluid

To identify the cell types in cultured amniotic fluid from the second trimester, we employed the single-cell transcriptome profiling of five cultured amniotic fluid samples: four from fetuses with spina bifida aperta (SBA) and one from a healthy fetus (Figure 1A). We used AF from fetuses with SBA, having access to AF from surgeries for the prenatal repair of the defect. In contrast, access to AF samples from healthy fetuses was limited due to the increase in non-invasive prenatal testing (NIPT) that substitutes amniocentesis in many cases [49]. AFC cultures at passage 0 (P0) showed a heterogeneous morphology, consisting of various cell types growing in colonies or as individual cells (Appendix A). The AF sample information is summarized in the Appendix A. A total of 43,687 cells were used in the analysis after removing empty droplets, doublets and low-quality cells. On the other hand, we kept 11,614 genes which had at least 1 UMI count in at least 0.01 of these cells. Using the 3000 most variable genes on the merged dataset, unsupervised clustering identified 14 major clusters (Figure 1B), in which all AF samples were represented (Appendix A).

The five most significantly differentially expressed genes for each cluster are shown in Figure 1C. Based on differential gene expression analysis and expression of cell-type-specific markers, the clusters were identified to be of the following tissue origins: renal (clusters 0, 1, 2, 5, 6, 7 and 9), placental (clusters 3, 11 and 13), neural (clusters 4 and 12) and immune (cluster 10) origin (Figure 1B). Cluster 8 was removed from further analyses, since it could not be identified with any cell type and it contained only low-quality cells (Appendix A). Nine different cell types were identified: renal tubular epithelial cells (RTEC, 41% of total cells, clusters 1, 2, 6 and 7), nephron progenitor cells (NPC, 16.1% of total cells, cluster 0), placental stromal cells (PSC, 10.5% of total cells, clusters 3 and 13), neurons (N, 9.4% of total cells, cluster 4), renal smooth muscle cells (RSMC, 7.3% of total cells, cluster 5), renal stromal progenitor cells (RSPC, 3.5% of total cells, cluster 9), immune cells (IC, 3.5% of total cells, cluster 10), syncytiotrophoblasts (SCT, 2.9% of total cells, cluster 11) and neuroglia (NG, 1.9% of total cells, cluster 12) (Figure 1B,D,E). The differentially expressed genes for each cluster are listed in the Appendix A. All identified cell types were present in all analyzed AF samples, with the exception of the neural-derived cells of clusters 4 and 12, which were predominantly detected in AF samples from fetuses with SBA but were barely present in the AF sample from the healthy fetus (see Results section on neural clusters).

The expression of the Y-chromosome gene *RPS4Y1* (male-specific) was used to determine if the clusters were of fetal or maternal origin. In the merged dataset of all five AF samples, which includes cells from two male and three female fetuses (Appendix A), expression of the gene was observed in all clusters, suggesting that all clusters were of fetal origin (Figure 2A). When investigating each AF sample individually, the *RPS4Y1* gene was detected in all clusters in AF from the two male fetuses but was absent in all clusters of samples from the female fetuses (Figure 2B–F), confirming that this approach can accurately identify the maternal or fetal origin of the cells.

The immune cell cluster (cluster 10) appeared to contain cells of macrophage phenotype, based on the macrophage markers *CD14*, *MSR1*, *MRC1* and *CTSS* (Figure 1C and Appendix A). All other clusters will be discussed in detail in the following sections.

### 3.2. Clusters of Renal Origin Comprise Four Different Renal Cell Types

Cells of renal origin constituted 68% of the total cells found in AF cultures at P0. The renal clusters were identified by the expression of the nephric lineage marker *PAX8* [50] and renal-cell-type-specific genes (Figure 3A). We annotated cells of cluster 0 as NPC based on the expression of the nephron progenitor markers *LYPD1* [51,52] and *SIX1* [51,53]. These cells also express genes involved in kidney development and podocyte maturation, namely, *FOXC2* [54,55] and *TCF21* [56]. Enriched GO terms in the NPC cluster include stem cell differentiation confirming the progenitor nature of these cells, and processes associated with nephron progenitor metabolism [57] (Figure 3B).

The most abundant cell type among the renal clusters was the RTEC. Our analysis identified four clusters of RTEC (clusters 1, 2, 6 and 7), which expressed the nephric-lineage and tubule-associated marker *PAX2* [58,59], along with several genes implicated in tubular basement membrane maintenance and function, such as *FXYD2* [60,61] and *TINAG* [62,63,64]. Differentially expressed genes also include potassium channel subunits expressed by tubular cells (e.g., *KCNJ15* [65]), junctional markers (e.g., *PATJ* [66,67]), genes associated with renal tubule morphogenesis and cell differentiation such as *TFAP2B* [54,68], *BICC1* [69,70], *UGT2B7* [71] and *MTSS1* [72], and the cilia-associated marker *DCDC2* [73]. In addition to the development of a polarized epithelium, GO terms analysis also showed enrichment in metabolic pathways associated with tubular cell metabolism [74]. Cluster 7 consisted of proliferating RTECs, based on the expression of H1 linker histones [75], such as *HIST1H4C* (Figure 2B), and on the enriched GO biological processes related to mitosis (Figure 3C).

Cells in cluster 5 were identified as RSMC due to the high expression of several smooth muscle cell markers [76], such as *ACTA2*, *CNN1*, *MYL9* and *TAGLN* (Figure 3A), along with extracellular matrix genes (*COL3A1* and *COL6A3*). GO enrichment analysis supported the mesenchymal and contractile nature of the cells (Figure 3B) and corroborated our annotation of cells in cluster 5 as RSMC.

The fourth cell type of renal origin identified in cultured AF was RSPC. Cluster 9 annotation was based on the expression of several genes known as markers of the renal mesenchyme in the developing kidney, namely, *PAX3* [77], *NR2F1* [78] and *TBX3* [79] (Figure 3A). Cells in cluster 9 also expressed stromal markers, such as *COL13A1* (Figure 3A). GO terms analysis, however, did not show any significantly enriched biological process.

### 3.3. Neurons and Neuroglia Exist in AF Cultures from Fetuses with Spina Bifida Aperta

Clusters 4 and 12 were designated of neural origin based on the expression of the neural progenitor markers *NES* [80], *SOX2* [81] and *FABP7* [82,83], the CNS-related cell adhesion genes *NRCAM* [84] and *NCAM1* [85], and the neural markers *PCSK1N* [86], *KIF5C* [87] and *CKB* [88] (Figure 4A,B).

Specifically, cells in cluster 4 expressed genes associated with neuronal development, such as *GAP43* [89], *DRAXIN* [90], *CEND1* [91] and *EEF1A2* [92], as well as the mature neuron markers *NEFL* and *NEFM* [93]. On the other hand, gene expression in cluster 12 coincided with the transcriptional profile of neuroglial cells. In particular, marker genes of cluster 12 included the glial marker *MAP2* [94], genes related to astrocytes, such as *S100B* [95], *CRYAB* [96] and *PMP2* [97], and oligodendrocytes, such as *PTPRZ1* [98] and *C1QL1* [99] (Figure 4A,B).

Interestingly, all the above-mentioned CNS-associated genes were predominantly expressed in the AF samples from SBA, but absent in AF cells from the healthy fetus (Figure 4C,D). While some cells of the normal AF sample fell into the neural clusters 4 and 12 (Appendix A), they did not appear to be of neural origin, as they did not express neural lineage markers (Figure 4A,D). Among the SBA samples, SBA2, SBA3 and SBA4 had a stronger expression of the neural lineage marker set compared to SBA1 (Figure 4C). However, the SBA1 sample still differed from the normal sample in the expression of the CNS-associated genes *NRCAM*, *NCAM1*, *PCSK1N*, *EEF1A2* and *CRYAB*. Collectively, the neural marker expression data suggest that neurons and neuroglia existed only in the SBA samples. These cells presumably originated from the exposed spinal cord in AF due to the SBA lesion. This is in line with data from the literature which indicate that neural tube defects in the fetus alter the composition of the AF cell pool by shedding cells of neural origin in AF [11,100,101,102].

### 3.4. Clusters of Placental Origin Comprise Cells of Epithelial and Mesenchymal Phenotypes

Clusters 11, 3 and 13 were identified to be of placental origin based on their differential gene expression. Cells in cluster 11 expressed several trophoblast-associated genes [103,104], such as *KRT7*, *GATA3*, *HAND1* and *TFAP2A* (Figure 5A). They appeared to have an epithelial phenotype, expressing markers of stratified epithelial cells (*EPCAM*, *SFN* and *DMKN*), junctional proteins (*JUP* [105], *TACSTD2* [106] and *CLDN7* [106]) and cell-adhesion molecules (*LAMA3* and *CDH1*) (Figure 5A). Based on the above and on the expression of the syncytiotrophoblast markers *KISS1* [107] and *SPINT1* [108] (Figure 5A), cells in cluster 11 were identified as SCT. These cells highly expressed cytokeratin genes, such as *KRT8*, *KRT18*, *KRT17* and *KRT19* (Figure 5B), which have also been described as trophoblast markers [109,110,111]. Using GO enrichment analysis of the gene sets representing each of the placental cell clusters, we observed that cells in cluster 11 demonstrated gene sets associated with epithelial cell phenotype and placenta development (Figure 5C).

On the other hand, cells in clusters 3 and 13 expressed the placenta-associated genes *FLT1* [112], *PAPPA* [113], *TIMP3* [114], *TWIST1* [115] and *HAND2* [116] (Figure 5A), and had a stromal phenotype. They expressed fibroblast markers, including *DDR2* [117], *FAP*, *S100A4* [118], *METRNL* [119] and *THY1*, and the gene *EMILIN1* [120], associated with the placental stroma. They also expressed markers related to extracellular matrix production by mesenchymal cells, such as *HAS2* [121] and *PCOLCE* [122] and the collagen genes *COL6A3* and *COL1A2* (Figure 5A). Enriched GO terms corroborated the stromal origin of these cells, highlighting biological processes related to extracellular matrix organization, mesenchymal cell differentiation and collagen-processing pathways (Figure 5C).

The expression of epithelial (cytokeratins, EpCAM) and stromal markers (CD90) was confirmed by immunofluorescence on fixed AF cell cultures at P0 (Figure 5D) and on cytospinned preparations of detached AF cells at P0 (Figure 5E).

### 3.5. Investigation of Stem Cell Marker Expression in Cultured AF from Four Fetuses with SBA and one Healthy Fetus

The presence of AF cell populations that share features with pluripotent stem cells has been suggested based on published evidence on the expression of pluripotency markers, such as Oct-4 [15] and Nanog [17], and the hematopoietic and embryonic stem cell marker c-Kit [21,22]. However, more recently, the expression of various pluripotency markers (e.g., Oct-4 and Nanog) in AF cells has been called into question [28]. To elucidate the presence of embryonic stem-cell-like populations in the AF samples of our study, we investigated the expression of the pluripotency marker genes *POU5F1*, *KIT*, *SOX2*, *NANOG* and *KLF4* on the transcriptome level in our scRNA-seq dataset. The genes *KIT* and *NANOG* were filtered out due to low expression and were not detected in our dataset (we removed genes that did not achieve at least 1 UMI count in at least 0.01 fraction of the cells that passed the QC filtering). *POU5F1* expression was restricted in a very limited number of cells which were spread among several clusters and, therefore, cell types (Figure 6A). *SOX2* was highly expressed in neural clusters 4 and 12 (Figure 6A), and this can be explained by its additional function as a marker of neural progenitors [123]. Finally, *KLF4* was among the top differentially expressed markers in cluster 11 (SCT, Figure 6A), having a role in epithelial cell proliferation and differentiation [124]. However, none of the clusters could be designated as embryonic-stem-cell-like based on the expression of the above pluripotency markers, as their expression did not coincide in any of the clusters and was overall limited.

c-Kit is a transmembrane receptor expressed predominantly in hematopoietic stem cells, and has been proposed for the isolation of AF cells with stem-cell-like properties [21,22]. After not detecting *KIT* in our scRNA-seq analysis due to low expression levels, we sought to investigate the expression at the protein level in cultured AF cells. AF cells from fetuses with SBA at P0 (different AF samples than those used for scRNA-seq, gestational age 23–26 weeks) were stained and sorted for c-Kit by fluorescence-activated cell sorting (FACS) or magnetic-activated cell sorting (MACS). By FACS, approximately 4% of the cells at P0 appeared to be c-Kit-positive, on average from four different experiments (example plot in Figure 6B and all FACS data in Appendix A). However, the cells did not grow in culture after sorting. MACS sorting was also tested, as an alternative to FACS with potentially improved cell viability outcomes [22]. Contrary to FACS, cells did grow in culture after MACS sorting; however, the purity of the isolated cells was low, as shown by immunostaining of cytospinned cell preparations immediately after MACS sorting (Figure 6C).

In addition to the c-Kit-positive cells, the presence of mesenchymal stem cells (MSCs) has also been proposed in cultured AF [23,24]. MSCs express CD105, CD73, CD90, CD29 and CD44, and lack expression of CD34 and CD45 [125,126]. In the five AF samples of the scRNA-seq, we investigated the expression of the MSC markers *NT5E* (CD73), *ENG* (CD105), *ITGB1* (CD29), *CD44* and *THY1* (CD90) at the transcriptome level. Further, we examined the expression of these markers at the protein level with immunostaining for flow cytometry using two amniotic fluid samples from fetuses with SBA (additional AF samples to those used for scRNA-seq, gestational age 23–26 weeks). We found that *NT5E*, *ITGB1* and *CD44* were expressed in all clusters (Figure 6D). Flow cytometry analysis corroborated the expression of CD29 and CD44 by all cells at the protein level, but CD73 was not detected by flow cytometry (Figure 6E), in contrast to the scRNA-seq data. On the other hand, *ENG* was highly expressed only in the macrophage cluster and moderately expressed in the stromal clusters RSMC, PSC1 and PSC2 (Figure 6D). The overall low level of expression of CD105 was observed also at the protein level (Figure 6E). Finally, *THY1* was expressed predominantly in the placental stromal clusters PSC1 and PSC2 and also present in RSMC, RSPC, NPC, N and NG (Figure 6D), while it was absent in the epithelial clusters. This was also reflected at the protein level, where CD90 was detected in a percentage of the cells (Figure 6E). In summary, the only clusters expressing all typical MSC markers at the transcriptome level were PSC1 and PSC2. These cells were identified above as stromal cells derived from the placenta, and presented features associated with fibroblast phenotype and extracellular matrix remodeling. However, whether these cells are multipotent mesenchymal stromal cells or whether expression of these markers is associated merely with their stromal/mesenchymal phenotype is unclear.

## 4. Discussion

scRNA-seq is a powerful tool for studying gene expression at a single-cell resolution and for identifying cell lineages in a heterogeneous population. In this study, we employed single-cell transcriptomics in four samples from fetuses with spina bifida and one from a healthy fetus to elucidate the tissue origin and lineage of cells in cultured AF of 20–24-week gestational age. For all experiments, we used cultured AF cells at P0 to avoid the overgrowth of certain cell types over others with cell passaging, and thus reflect as accurately as possible the composition and relative numbers of viable, plastic-adherent cells in AF.

Our single-cell approach sheds light on the heterogeneity and composition of AF cells by overcoming the challenges of marker identification using bulk approaches (e.g., bulk sequencing and proteomics [16,31,37]). The transcriptomic profiling in our analysis revealed the major fetal tissue origins of cultured AF cells, namely, the urogenital tract and the placenta. An immune cell population with macrophage-relevant marker expression was also detected. In addition to those, cells of neural origin appeared in AF from fetuses with SBA, seemingly deriving from the exposed spinal cord as a result of the neural tube defect. Single-cell profiling detected 13 clusters of cells which were assigned to nine different cell types, based on marker expression. This finding refutes the theory that only two (spindle-shaped and round-shaped [28]) or three (epithelioid, fibroblastic and amniotic [12,33]) cell types exist in cultured AF. Our study also demonstrates that AF cells with epithelial or mesenchymal phenotypes can be derived from more than one fetal tissue; for instance, cells of both placental and renal origin include subtypes of epithelial and of mesenchymal phenotypes, which represent different cell lineages and express distinct markers according to their tissue origin. This study showed that even the so-called “mesenchymal” or “epithelial” AF cell populations may be, in fact, heterogenous.

The vast majority of adherent AF cells in our dataset appeared to be of renal origin. This is not surprising given that, in the second trimester, AF consists mainly of fetal urine [1]. It is known that urine contains large amounts of exfoliated viable cells of renal origin [127,128], such as tubular epithelial cells, renal progenitors and podocytes, which can be expanded in culture. Our data revealed the presence of four distinct cell types of renal origin in AF, namely, differentiated tubular epithelial cells, stromal cells, smooth muscle cells and nephron progenitors. The presence of a high number of nephron progenitors in our samples can be explained by the fact that nephrogenesis is not yet complete at a gestational age of 20–24 weeks (nephrogenesis is completed at approximately 34 weeks of gestation [127]). The existence of progenitor cells of renal origin in AF is of great interest and has been shown before for podocyte precursors [129] and renal mesenchymal cells [130]. The AF nephron progenitors, and all the AF cell types of renal origin identified here, have high potential for use in biomedical applications, such as in tissue engineering for kidney repair [127,128], kidney disease modeling and the study of fetal development [128]. Our work provides a list of markers for the identification and deeper characterization of these cells, which will aid to harness the full potential of AF cells for such purposes.

Four out of five AF samples used in the scRNA-seq analysis were from fetuses with SBA. This congenital defect of the central nervous system develops when the neural tube fails to close in early gestation, leading to the progressive injury of the exposed spinal cord and subsequent deterioration of the fetal neurological function. Our data showed that all samples from fetuses with SBA contained cells of neural origin, which were not present in the normal AF sample. These cells expressed the neural progenitor genes *SOX2* (except for sample SBA1) and *NES*, but also advanced differentiation markers of neurons and neuroglia. A limitation of this study is that it includes only one AF sample from a healthy fetus versus four samples from fetuses with SBA. However, our claim that it is the neural tube defect that contributes cells of neural origin to the AF cell pool is supported by both the nature of the pathology and published experimental evidence. On the one hand, in SBA, the spinal cord is exposed to the external environment during gestation, i.e., to AF, due to the failure of neural tube closure. The injury to the spinal cord is a consequence of the chemical damage from the exposure to AF, as well as of the mechanical damage within the uterus [102]. Therefore, it is conceivable that cells are exfoliated from the spinal cord and end up in AF. On the other hand, neural tube defects have been shown experimentally to shed neural cells in AF [11,100,101,102]. Our findings build upon these published observations. The comprehensive list of markers that we provide in our scRNA-seq dataset can help understand the progression of SBA, and identify potential targets and diagnostic tools for the disease. For instance, it can aid in the elucidation of the pathophysiological mechanisms involved in spinal cord injury, with the example of astrocytosis that is known to occur in SBA lesions [131]. Marker expression in our analysis confirmed this finding with the expression of various astrocyte-related markers in the neuroglial cell cluster. In addition, our study provides a better characterization of the neural-derived cell types found in AF from fetuses with SBA, paving the way towards the development of novel strategies for the prenatal diagnosis and repair of SBA, as well as new tissue engineering approaches for spinal cord regeneration for the treatment of neural tube defects [132].

Specifically for tissue engineering applications, the presence of stem cell populations in cultured AF is of great interest. Here, we investigated the presence of stem-like cells in cultured AF based on marker expression. The pluripotency markers *POU5F1* (encoding Oct-4) and *NANOG* were not detected in our analysis at a transcriptome level, in contrast to what has been published before. In addition, *SOX2* and *KLF4* were expressed in specific clusters but did not mark stem cell populations. Our data support the recently published findings that cultured AF cells of a gestational age of 20–24 weeks do not express Oct-4 and Nanog [28] under standard culture conditions (i.e., not in pluripotency-supporting conditions). On the other hand, the pluripotency marker c-Kit, which has been suggested for the isolation of multipotent AF cells, was detected at the protein level in our study, but not in the transcriptome analysis. The lack of *KIT* expression in our analysis might be explained by the higher gestational age of the AF samples compared to those in reports describing c-Kit-positive AF cells. Studies with c-Kit-positive isolated cells utilized cultured AF samples from amniocenteses [19,20,21,22], which are typically performed between 14 and 20 weeks of gestation, while the AF samples of this study were at 24 weeks for the SBA samples and at 20 weeks for the normal sample. Nevertheless, it has also been shown that the number of c-Kit-positive AF cells roughly follows a Gaussian distribution during gestation in humans, and peaks at 20 weeks [19,20]. Based on this observation, the number of c-Kit-positive cells should not differ considerably between samples of 14–20 and of 20–24 weeks. Therefore, whether the absence of *KIT* expression in our study is due to the higher gestational age of the AF samples is unclear. Regarding the presence of MSC markers in this study, they were unanimously detected at the transcriptome level only in the placental stromal cell clusters, but not all of them were found at the protein level. Whether the placental stromal cells are also multipotent MSC requires further investigation. In addition, progenitor-like cells are potentially more prominent in AF of lower gestational age compared to that of the AF samples analyzed in this work.

### Limitations of This Study

Several crucial aspects of our study must be pointed out. One limitation is that it includes only five AF samples. Although all identified cell types in our analysis were detected in all five samples, suggesting that all cultured AF cell types at this gestational age were identified, these findings need to be validated with a larger sample size. Another important consideration is that four out of the five samples used for scRNA-seq were from fetuses with neural tube defects, while only one sample was from a healthy fetus. It is, however, conceivable that the pathology does not affect the presence of other cell types, such as stem cells, but it only increases the number of adherent cells due to the contribution of neural cells to the cell pool [11,100,101,102]. Indeed, except for the neural cells, all other identified cell types were detected in all five AF samples, regardless of the presence or absence of the pathology. In addition, the gestational age of the healthy AF sample was four weeks lower than that of the SBA samples (20 weeks versus 24 weeks, respectively); however, the same cell types (except neural cells) were identified in the sample of 20 weeks and in those of 24 weeks, suggesting that these four weeks of difference at this gestational age do not influence the adherent cell types that can be found in AF. Further, we should highlight that in this study, we used the typical culture conditions for prenatal cell testing from amniocentesis samples, and for the isolation of c-Kit-positive cells, according to the literature; however, it is possible that different cell types will be detected in AF cells grown in different isolation media and culture conditions. Another essential consideration is that the total number of viable, adherent AF cells and the relative AF cellular composition may vary widely among subjects, and also in the same subject at different gestation ages (for instance, between the first and the third trimester). Finally, it is noteworthy that in addition to the adherent cells, AF potentially contains also viable, non-adherent cells that may be of interest but have not been investigated in this work.

## 5. Conclusions

In summary, this is the first report characterizing all cell types in cultured AF at the single-cell level and revealing the vast heterogeneity in the adherent cell populations. Our study provides a comprehensive gene expression atlas of cultured AF cell types based on scRNA-seq. Despite the lack of cells with pluripotency features that we observed, our characterization of the diverse pool of AF cells will facilitate new approaches to tissue engineering and regenerative medicine applications. This new insight into AF cell phenotype and tissue origin will open up a range of possibilities for future research, from the study of fetal development in physiology and disease to the use of AF cell types in biomedical research and therapeutic applications.

## Figures and Tables

**Figure 1 cells-12-01577-f001:**
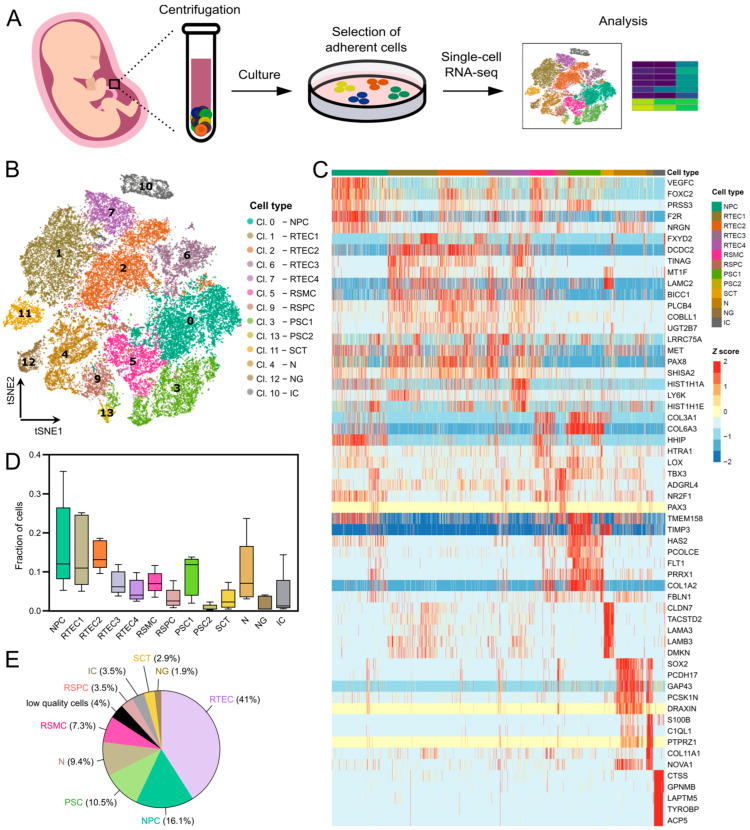
Global analysis of single-cell expression profiles in cultured human AF cells from the second trimester. (**A**) Overview of the experimental and bioinformatics analysis workflow. Human AF samples of 20–24-week gestational age were cultured and detached at passage 0 for scRNAseq. (**B**) Unsupervised clustering and t-SNE plot of 45,516 single cells from five human AF cultures (AF from four fetuses with spina bifida aperta (SBA) and one from a healthy fetus). The 13 identified cell clusters are shown in different colors and the labels indicate the assigned cell type. (**C**) Heatmap with the expression pattern of the top five cluster-specific genes in the 13 clusters. The *Z* score indicates relative expression. (**D**) Box plots representing the contribution of the different AF samples (n = 5) to each cell cluster. (**E**) Pie chart representing the percentage of each cell type detected in the five AF samples.

**Figure 2 cells-12-01577-f002:**
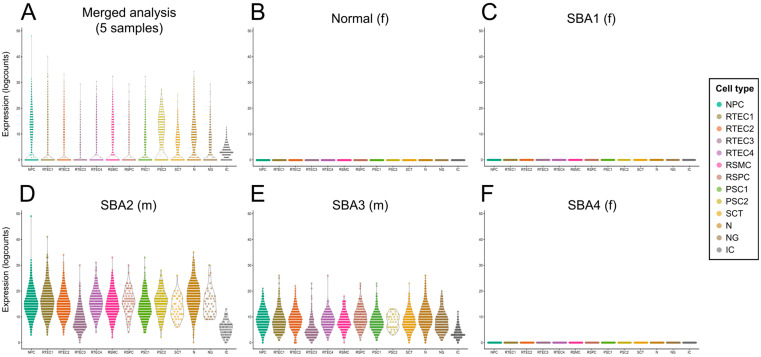
Identification of fetal/maternal origin of the cell types detected in the cultured AF samples of this study. (**A**) Violin plot showing the expression of the male-specific gene *RPS4Y1* per cluster in the merged analysis of the five AF samples included in the scRNA-seq analysis (two from male and three from female fetuses). The gene was expressed in all clusters, indicating that all detected cell types were of fetal origin (data from AF samples from four fetuses with spina bifida aperta (SBA) and one from a healthy fetus). (**B**–**F**) Violin plots showing the expression levels of *RPS4Y1* per cluster in each of the five AF samples included in the scRNA-seq analysis. The gene was expressed in all clusters in the AF samples from the two male fetuses (m) but was absent in all clusters in the AF samples from the three female fetuses (f), indicating that all detected clusters (and therefore cell types) were of fetal origin. The cell clusters are shown in different colors and the legend indicates the assigned cell type.

**Figure 3 cells-12-01577-f003:**
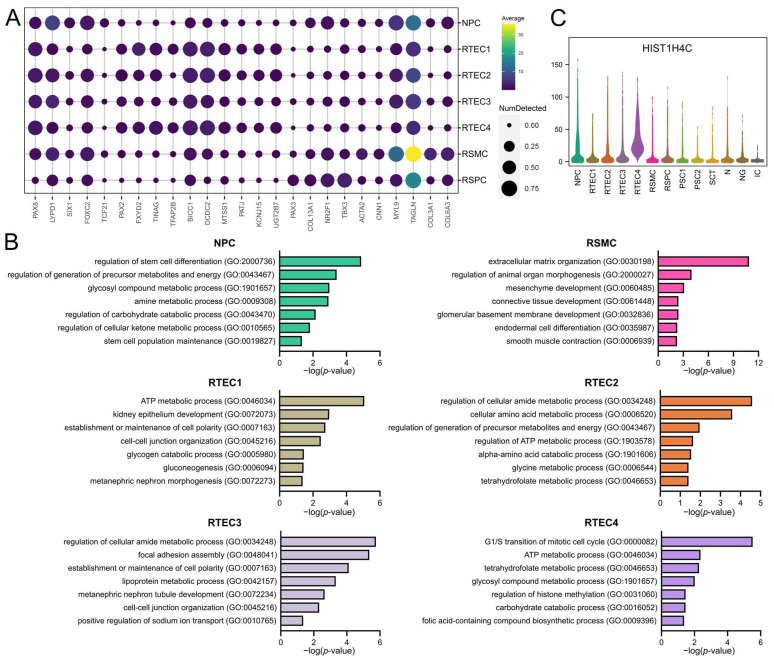
Characterization of AF cell types of renal origin. (**A**) Dot plot showing the expression of a selection of known nephric-lineage markers and cell-type-specific genes in subsets of AF cells of renal origin. (**B**) Biological processes enriched in each AF cluster of renal origin, as identified by Gene Ontology enrichment analysis with the differentially expressed genes. GO terms are depicted and ranked by adjusted *p*-value (<0.05). The RSPC cluster did not show any significantly enriched biological process. (**C**) Violin plot showing expression of the gene *HIST1H4C* among all clusters. Expression was elevated in RTEC4, suggesting that this cluster contained cells that undergo mitosis.

**Figure 4 cells-12-01577-f004:**
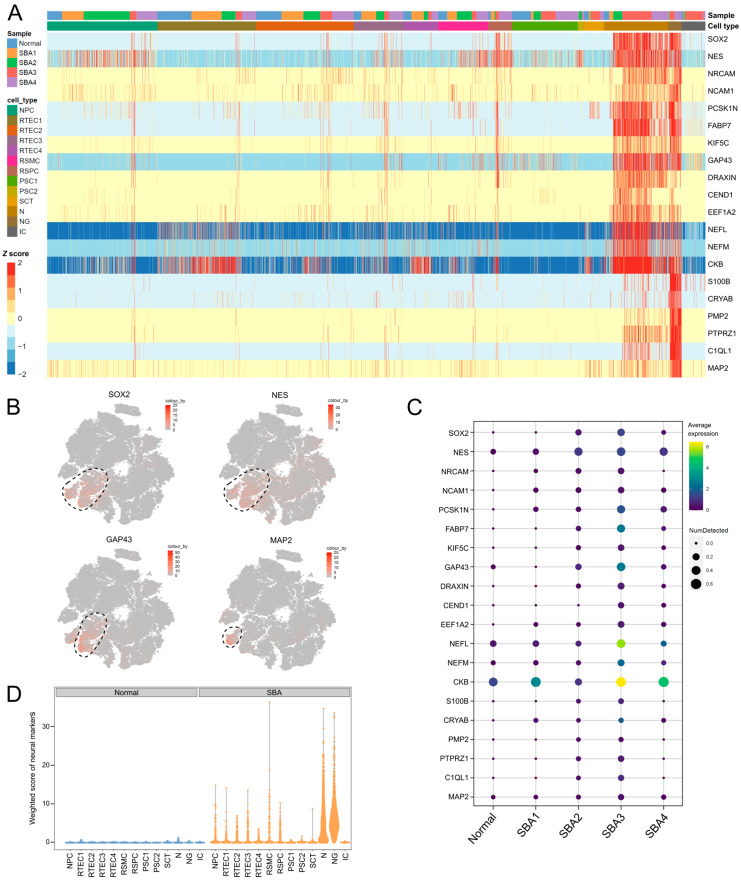
Neural-associated marker expression in one normal AF and in four AF from fetuses with neural tube defects. (**A**) Heatmap with the expression pattern of neural-cell-lineage markers and selected neuron- and neuroglia-associated markers in the 13 clusters and in each of the five AF samples that were subjected to scRNA-seq. The *Z* score indicates relative expression. (**B**) t-SNE plots of all single cells with the expression of representative neural-cell-lineage-specific marker genes. *SOX2* and *NES* for neural progenitors, *GAP43* for neurons and *MAP2* for glial cells. Black dashed lines mark the neural clusters of interest. (**C**) Dot plot highlighting the expression of the neural-associated genes from (**A**) in each of the five AF samples included in the scRNA-seq analysis (one AF sample from a normal fetus and four from fetuses with SBA). (**D**) Weighted mean expression score of the neural-cell-lineage marker set from (**A**) in the one normal versus the four SBA samples. The score was computed by the package *decoupleR*.

**Figure 5 cells-12-01577-f005:**
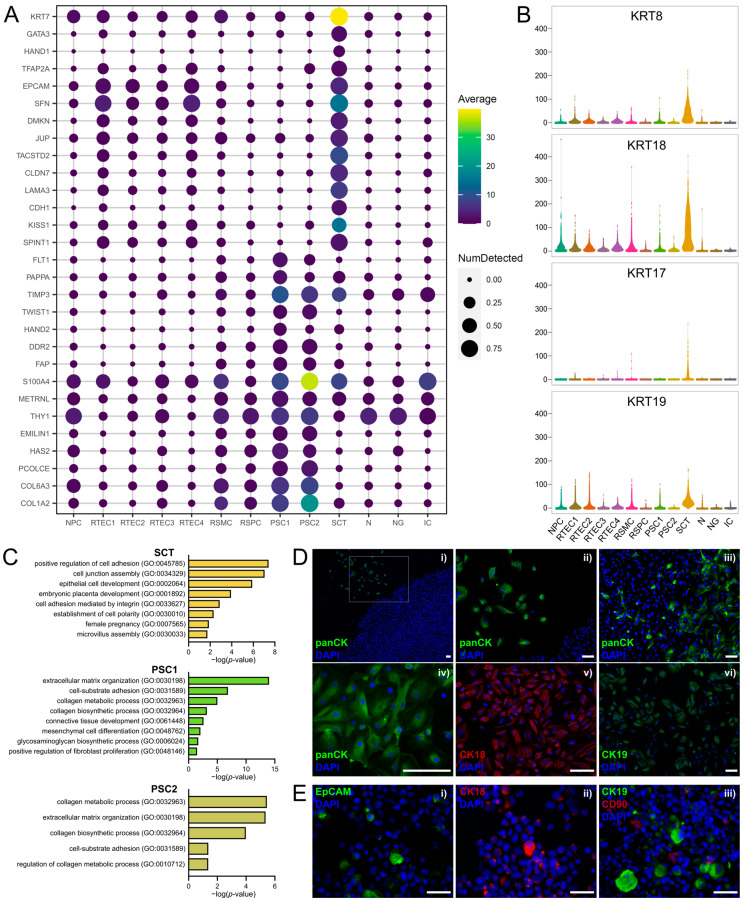
Cell-type-specific gene and protein expression in the clusters derived from the placenta. (**A**) Dot plot showing the expression of selected placental-cell-type-specific genes. (**B**) Violin plots with the expression of cytokeratin genes in all clusters, highlighting the high expression of cytokeratins in the epithelial cluster SCT. (**C**) Biological processes enriched in each AF cluster of placental origin, as identified by Gene Ontology enrichment analysis with the differentially expressed genes. GO terms are depicted and ranked by adjusted *p*-value (<0.05). (**D**) Immunofluorescence staining of cytokeratins in adherent AF cultures from fetuses with SBA at P0. (i)–(iv) Detection of cytokeratins with a pan-cytokeratin antibody. (i)–(iii) Images including positive and negative populations for cytokeratins in cultured AF at P0. Panel (ii) shows a magnification of the boxed area depicted in panel (i). (iv) AF cell population positively stained with the pan-cytokeratin antibody. (v) AF cell population positive for cytokeratin 18. (vi) AF cell population positive for cytokeratin 19. Nuclei were stained with DAPI (blue). Scale bars: 100 µm. (**E**) Immunofluorescence staining for EpCAM (i), cytokeratin 18 (ii) and co-staining of cytokeratin 19 and CD90 (iii) in cytospinned preparations of detached AF cultures from fetuses with SBA at P0. Nuclei were stained with DAPI (blue). Scale bars: 50 µm.

**Figure 6 cells-12-01577-f006:**
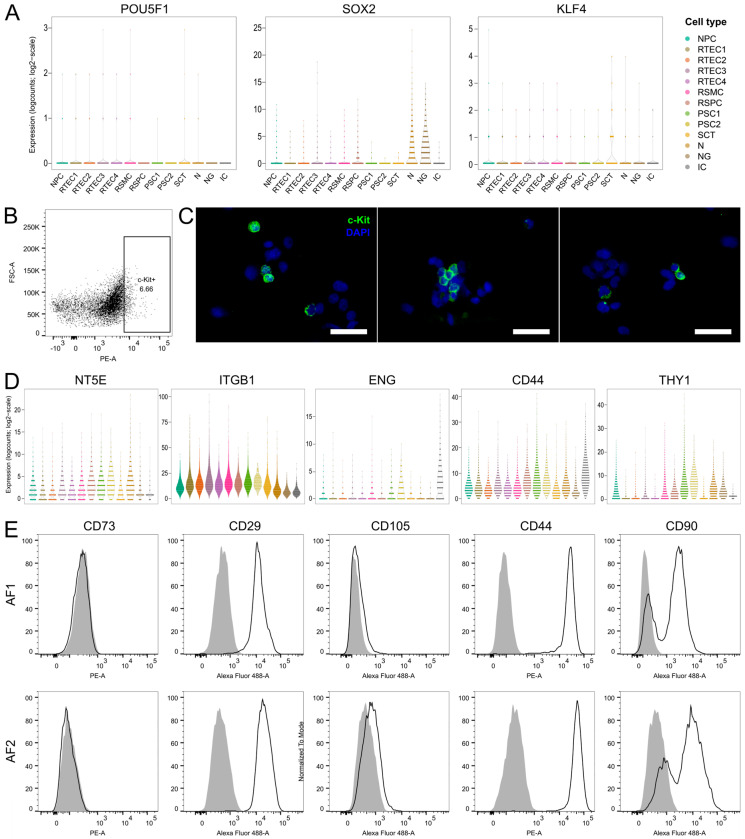
Expression of pluripotency markers and mesenchymal stem cell markers in cultured AF cells. (**A**) Violin plots showing the expression of the pluripotency markers *POU5F1*, *SOX2* and *KLF4* in all clusters. *POU5F1* was expressed in very low levels in all clusters, *SOX2* was expressed in the neural-derived clusters and *KLF4* was expressed in the epithelial cluster SCT. (**B**) Expression of c-Kit in cultured AF cells from fetuses with SBA at P0. Representative FACS plot from one of the four tested AF samples. FACS plots of all tested AF samples are shown in Appendix A. The gate for c-Kit-positive cells was set using the isotype control antibody. The same gating was used for the cell sorting. FACS-sorted c-Kit-positive cells did not grow in culture. (**C**) Immunofluorescence staining of c-Kit in MACS c-Kit-selected AF cells (P0) after sorting. The purity of c-Kit-positive cell population post-selection was low, as shown by the presence of many c-Kit-negative cells. Nuclei were stained with DAPI (blue). Scale bars: 50 µm. (**D**) Violin plots depicting the expression levels of the mesenchymal stem cell markers *NT5E*, *ITGB1*, *ENG*, *CD44* and *THY1* in all clusters. Cell type coloring is the same as in (**A**). (**E**) FACS histograms showing the expression of the mesenchymal stem cell markers from (**D**) at protein level in two AF samples from fetuses with SBA (AF1 and AF2). Gray filled histograms: isotype control antibody, black open histograms: marker staining as indicated in each plot.

## Data Availability

Our data have been deposited in the NCBI Gene Expression Omnibus (GEO) under accession number GSE206696.

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
