# Peer review of "scRNA-Seq of Cultured Human Amniotic Fluid from Fetuses with Spina Bifida Reveals the Origin and Heterogeneity of the Cellular Content"

_cells, 2023, doi:10.3390/cells12121577_

Round 1

Reviewer 1 Report

In this manuscript, Desargyri and co-Authors adress a very interesting topic in the characterization of human amniotic fluid stromal cells at single cell level by RNAseq. Nevertheless, I am afraid that Authors should carefully reconsider the main focus of their study (and the title itself). Indeed, they aim at defining tissue origin and marker expression of amniotic fluid stromal cells, but they rather present data referring only to pathological specimens, i.e. about cells from amniotic fluid retrieved during surgical intervention for spina bifida. Their results should be carefully and specifically discussed in such perspective.

An n=1 as healthy control is not acceptable. I do understand that it may be difficult to find healthy controls matching the same gestational stage at which the pathological ones have been obtained; yet, Authors should try to get healthy samples from amnio-reduction procedures.

More in general, an n=5 samples as used for RNAseq is again quite debatable and it does not allow to infer meaningful results for publication. 

There is no information about the donors (i.e. age, health condition etc).

When comparing and discussing their results with data already published in literature about human amniotic fluid-derived MSC or progenitor cells (such as the c-Kit+ ones)Authors should very carefully consider that the gestational age at which their cells have been retrieved is completely different. Indeed most studies refer to amniocentesis-derived samples to obtain cells from, that is specimens obtained at 14-16 weeks gestation, significantly earlier than the samples considered in this study. Moreover, the earlier the stage at which human amniotic fluid cells are obtained from, the more likely they can still express immature and progenitor-like properties.

Resolution of the immunostaining pictures of the cells should be improved. Likewise, Authors should also provide bright field images of cultured cells.

In light of these observations, I strongly suggest Authors to significantly revise their study and consider major revision for the manuscript to meet the standards of Cells journal.

Reviewer 2 Report

I would like to thank the editor for giving us the opportunity of revising this piece of work from Dasargyri and colleagues. The article describes in detail the single cell transcriptomic profile of cultured amniotic fluid cells at their earliest passage (P0), which has a great value for the field. As of today, there is no reports available in the literature aiming at addressing the proposed research question using bulk nor single cell sequencing approaches. This makes the work novel and interesting for the AF cell biology community. Overall, the authors provide good cluster definition, good markers selection and the spina bifida sample size is appropriate. However, the sample size of the control group (n=1!) is inappropriate, and not sufficient to support some of the conclusions presented. Especially considering the authors own discussion about the variability and heterogeneity of the amniotic fluid between patients or time points. The amniotic fluid is an easily accessible resource (i.e. diagnostic surplus), hence producing further control data should not have been problematic. In my opinion this is the biggest flaw of the article and can be addressed with revision. I present here a point-by-point list of concerns, that if addressed properly would, in my opinion, help improving the quality of the article and further support the interesting findings presented.

Title:

The title should be reviewed. The authors mention “second trimester” as a whole. While all their SB samples come from 24w pregnancies, and their single control comes from a 20w pregnancy. This window of gestation covers only about 1 month of gestation, and is not representative of the whole trimester. Moreover, there is a 1-month gap between the two experimental groups, which may be problematic for sample comparison to the control as time-dependent differences will be detected. I would suggest the authors to rephrase the title accordingly.

Introduction:

1 On page 2 line 62-64: “(AFMSCs)(22,23) has also been proposed, following the observation that mesenchymal cells exist among expanded AF cells(24)” The ref 24 utilised AF from ewes not human, and was published prior to 22 and 23. One or more human examples of AFMSC existence should be included, possibly following a temporal order.

2 On page 2 line 73-75:” Early classifications based on morphological and cell growth criteria suggested the presence of two (spindle-shaped and round-shaped(28)) or three (epithelioid, fibroblastic and amniotic(12,32)) cell types.” The ref 28- Vlahova F, 2019 distinguish two cell types based on morphology and immunophenotype. In my opinion this should be supplemented with Roubelakis, Maria G, 2011- the first paper describing RS and SS-AFMSC.

Results:

1 As discussed above, my biggest concern is with the presentation of one single healthy control sample. The authors make a big point about heterogeneity, but then assume this one sample is representative of all the cell clusters present in the healthy second trimester of gestation. This needs to be further supported by producing additional control data. Importantly, the lack of age-matched controls is understandable (20 vs 24) but needs pointing out in the discussion when they discuss limitations of this study. This is in line with the title amendment I suggested above.

2 The authors should clarify which precise kit of the 10x scRNAseq platform was run. What version, 3’ or 5’, single or dual index. Each of these approaches may lead to different capture rate for the various clusters, hence clarification is required.

3 Doublets detection has been run only through R on the count matrix using scDblFinder. For better QC, I strongly recommend also running doublet detection on the raw .bam files using a program such as souporcell.

4 It is uncommon for clustering resolution to be set below 0.5 without justification, how was the resolution of 0.4 chosen? Moreover, I am unsure about the clusters definition (i.e. part of 2 falling into 0, part of 3 falling into 13, par of 0 falling into 9). Can you please clarify your approach further?

5 What particular metric was used to judge Cluster 8 as low quality and remove it? It seems that all that was used to justify removal is a low number of total genes and low UMI. This could be explained by, for example quiescence, senescence or apoptosis processes. The authors should show more data on cluster 8, including mitochondrial gene %, expression of sex specific genes like RPS4Y1 (to exclude maternal origin), cell cycle genes and individual sample breakdown (from SB or healthy)

6 It is surprising to me that cKIT was not detected at a depth of 50k reads/cell. The authors explain this by stating cKIT+ cells do not adhere and are therefore lost before sequencing. However, this is inconsistent with a broad number of reports, first of which De Coppi et al Nature Biotechnology 2007, where Kit is used for the selection of cells before and after an initial culture step. Overall, this completely goes against the canonical use of KIT as an AFSC marker, and is in contrast with what has been observed by several different groups in the field. In addition to this, it is worth bearing in mind that all the strongest KIT expressing cells are hematopoietic stem cells (HSCs), which poorly attach to plastic, and have lower capture in 3’ sequencing. The authors should discuss this in more detail, and possibly highlight this in the limitation section of the discussion.

7 The authors’ data are presented as a novel disagreement with the presence of pluripotent AF stem cells, but the lack of pluripotency has been known for a while (take Vlahova et al. as an example). Consequently, this novelty claim on this should be toned down. They then say that whether the placental stromal cells are multipotent MSC needs further investigation, but actually this is one of the two most important questions approached in this article, with the second being the presence of spina bifida neuro- cells in the AF. It seems feasible and fair to request them to undertake this additional investigation into the multipotent MSCs.

8 Page 6 Line 281: “The differentially expressed genes for each cluster are listed in the Supplementary information (Table S2). – No table S2 in supplementary

9 Page 6 Line 282-285: according to authors argument and figure S1a & S1b; cluster 9, 10 and 13 also have cells predominantly from SBA samples than healthy samples. Further explanation needed, and further control data should be produced to support this strong claim. The authors should show differences between healthy and SB by presenting the data separately and indicating clearly which cells express neural markers in the CT. How do the authors explain the presence of those cells in the healthy sample?

10 It is not fully clear to me how many cells passed QC for each sample. This should be presented in the supplementary table 1. Moreover, the authors could show a tSNE coloured by sample to better visualise the possible distribution differences across different individuals. From Figure S1a, it seems that not many cells are from SBA1 in some clusters, while cells from normal and SBA2 samples are dominating the plot. While this does not seem to be true based on numbers of cells sequenced for each sample in Table S1. The authors could present more distinguishable colour combinations, like in S1b, and reduce the size of the dots in the tSNEs for a better data visualisation in S1a.

11 Page 9 Line 352-353: Although the authors pointed out no significantly enriched biological process for cluster 9 (RSPC), this GO enrichment analysis results should be reported in figure 3b as done with other 6 renal clusters.

12 P16 line 521-526, the authors demonstrated that cells from placenta and renal origins have mesenchymal and epithelial phenotypes. How do the authors explain the co-expression of epithelial and mesenchymal genes in the placental and renal clusters?

13 Figure 6b needs unstained CT / isotype. What antibody was used? Has this been tested on a positive CT? In Figure 6b and S3 the authors showed cKIT expression for 4 SBA samples despite not having detected Kit at transcriptomic level. A positive and negative controls should be added to the panel. In addition, figure 6e presented flow cytometry results of MSC makers for 2 SBA samples, the remaining 2 SBA AF could also be tested, but the normal AF cells must be included as healthy control.

Data Availability

The GEO accession number presented in the manuscript provided the raw data for each of the 5 samples. Labelling needs improving for the files, there is no consistency and no distinction between healthy and SBA in the title. However no metadata data is shown here. Moreover, there are also links for the SRA database shown, however this data was inaccessible to the reviewer. Detailed and extensive metadata is key in making this data open and reusable. For each cell, the metadata should include at least: Basic QC (Number of counts, number of features, % mitochondrial genes), doublet probabilities, any cell labels assigned and for completeness, the sample of origin and associated timepoint. Furthermore, the processed object should be made available with UMAP/TSNE reductions present so that the data endpoint is also accessible.

Discussion:

1 Page 17 line 547-550, the authors indicated “all samples from fetuses with SBA contained cells of neural origin, which were not present in the normal AF sample”. This is not convincing, as AF cells from normal sample do fall into cluster 4 and 12, annotated as neurons and neuroglia in Figure S1b. The authors might refer only to the CNS-associated genes NRCAM and NCAM1 that are expressed in all SBA but not normal AF samples (p10 line 375-376), but this evidence is not very strong in my opinion. In addition, as we have mentioned above, one single normal AF sample (n=1) is not enough to draw any conclusions.

2 Line 549 “These cells expressed the neural progenitor genes SOX2 and NES”, however SBA1 cells didn’t seem to express SOX2 in figure 4C. Moreover, the normal and SBA2 sample seemed to have similar expression of neural genes.

3 Page 17 line 580-582 the reference must be provided to support the statement “increase the number of adherent cells due to the contribution of neural cells to the cell pool”.

4 Overall, while I appreciate the innovative content of the work, I would like to see a more critical and thorough discussion of the constraints of this study

Round 2

Reviewer 1 Report

Authors addressed the comments that have been raised mainly by rebuttal. I still think that a major limit is represented by the low n number of the samples. If more samples are not accessible at all, Authors should then indicate that as a main limit in their study and explain the frequency and average of spina bifida samples they can usually expect. They can comment on that in a dedicated "limits of the study" sub-paragraph within the discussion at page 14 of the manuscript pdf file, since it is briefly mentioned as it is in the current state.

Reviewer 2 Report

The authors implemented my comments satisfactorily. 

Author Response

We thank the reviewer for the constructive feedback.